# Evolving Acquired Vemurafenib Resistance in a BRAF V600E Mutant Melanoma PDTX Model to Reveal New Potential Targets

**DOI:** 10.3390/cells12141919

**Published:** 2023-07-24

**Authors:** József Tóvári, Diána Vári-Mező, Sára Eszter Surguta, Andrea Ladányi, Attila Kigyós, Mihály Cserepes

**Affiliations:** 1Department of Experimental Pharmacology, National Institute of Oncology, 1122 Budapest, Hungary; 2National Tumor Biology Laboratory, National Institute of Oncology, 1122 Budapest, Hungary; 3Department of Surgical and Molecular Pathology, National Institute of Oncology, 1122 Budapest, Hungary; ladanyi.andrea@oncol.hu; 4KINETO Lab Ltd., 1037 Budapest, Hungary; info@kinetolab.hu

**Keywords:** melanoma, BRAF V600E, PDTX, acquired resistance, resistance evolution, preclinical resistance model, patient-derived tumor xenograft model

## Abstract

Malignant melanoma is challenging to treat, and metastatic cases need chemotherapy strategies. Targeted inhibition of commonly mutant BRAF V600E by inhibitors is efficient but eventually leads to resistance and progression in the vast majority of cases. Numerous studies investigated the mechanisms of resistance in melanoma cell lines, and an increasing number of in vivo or clinical data are accumulating. In most cases, bypassing BRAF and resulting reactivation of the MAPK signaling, as well as alternative PI3K-AKT signaling activation are reported. However, several unique changes were also shown. We developed and used a patient-derived tumor xenograft (PDTX) model to screen resistance evolution in mice in vivo, maintaining tumor heterogeneity. Our results showed no substantial activation of the canonical pathways; however, RNAseq and qPCR data revealed several altered genes, such as *GPR39*, *CD27*, *SLC15A3*, *IFI27*, *PDGFA*, and *ABCB1*. Surprisingly, p53 activity, leading to apoptotic cell death, was unchanged. The found biomarkers can confer resistance in a subset of melanoma patients via immune modulation, microenvironment changes, or drug elimination. Our resistance model can be further used in testing specific inhibitors that could be used in future drug development, and combination therapy testing that can overcome inhibitor resistance in melanoma.

## 1. Introduction

Malignant melanoma of the skin occurs in over 320,000 new patients annually, and accounts for over 57,000 deaths yearly, based on the GLOBOCAN estimates [1]. The potential effects of therapy strongly depend on the clinical stage of the disease at first diagnosis. Given that cutaneous melanomas can be easily eliminated surgically, the localized disease has about 99% of 5-year relative survival rate. On the other hand, the same rate for those with regional metastasis declines to 71% and with distant metastasis to 32% [2]. The numbers clearly show that the systemic therapeutic approaches need further improvement.

For high-risk melanoma, surgical excision is supplemented with adjuvant chemotherapy, while unresectable stage III–IV tumors can undergo immunotherapy (pembrolizumab, nivolumab, ipilimumab, relatlimab, IL-2), become targeted by signal-transduction inhibitors (BRAF inhibitor vemurafenib or dabrafenib, MEK inhibitor trametinib or cobimetinib, and KIT inhibitor imatinib mesylate), or classical chemotherapy agents (dacarbazine, temozolomide, cisplatin, vinblastine, carmustine, tamoxifen, and paclitaxel) [3]. In addition, stereotactic radiosurgery or external-beam radiation can be an option for some patients. A recent study shows that about 50% of melanoma patients carry a mutation of the BRAF protein, and roughly 75% of them (38% of all patients) received a BRAF inhibitor as a therapy [4].

The first BRAF inhibitor (BRAFi) vemurafenib was approved in 2011 [5]. BRAF inhibitors include vemurafenib, dabrafenib, and encorafenib, often applied in combination with MEK inhibitors trametinib, cobimetinib, or binimetinib in clinics. The inhibition of BRAF or MEK, as well as dual inhibition, proved to be successful in terms of antitumor effect [6]. Sadly, over time, acquired resistance of the cancer cells often arise, suppressing therapeutic efforts.

The literature on possible mechanisms of BRAFi resistance is excessive on cell line-based models. As summarized in recent review articles [7,8,9,10], the most abundantly identified processes in melanoma cells through BRAFi treatment were the following: BRAF (gene amplification, splicing, secondary mutations, dimerization), NRAS (mutation), MEK1/2 (mutation), receptor tyrosine kinases and their interaction proteins (upregulation), PI3K or AKT (mutation), STAG2-STAG3 (downregulation), YAP/TAZ pathway (activation), dual-specificity phosphatases (DUSPs; downregulation), RAC1 (mutation), NF1 (somatic mutation), RNF125 (downregulation), the DBL family of guanine exchange factors (GEFs; activating mutations). Many of these events confer activation of downstream MAPK pathway (or parallel pathways like PI3K-AKT, DBL/RAC1/PAK) independently of BRAF activity, therefore restoring cell proliferation and avoiding apoptotic cell death. The changes in BRAF bypass the BRAFi effect (BRAF dimers and upregulated expression lead to BRAFi resistance) and keep the signaling active.

These data proved useful in the clinical setup, as they lead to the introduction of combination therapies targeting BRAF and MEK1/2 simultaneously, improving melanoma prognosis [11,12]. However, these models represent isogenic, clonelike cancer cells, limiting the usability of the results in many real-life cases because of excluding many important cancer cell types (especially less attached, more stemlike cancer cells, which carry the skillset to develop clinical resistance), spatial heterogeneity, and tumor microenvironment [13].

Several studies attempted the use of PDTX models to track acquired BRAFi resistance. Kemper et al. reported a broad number of clinical samples from 89 patients, obtaining surgical sample from both treatment-naive and therapy-resistant tumors. They established and examined the PDTX tumors, finding that duplication of BRAF V600E kinase domain can confer to resistance, and can be reversed by pan-RAF dimerization inhibitor treatment in mice [14].

Another study investigated several PDTX models, finding that different models undergo different routes of change. Either *BRAF* alternative splicing, NRAS Q61 mutation, *MAP3K8* overexpression, and increased mutant *BRAF* gene amplification were identified, all mechanisms clinically identified from BRAFi-resistant patient samples [15].

In a clinical study of 45 patients [16], MAPK and PI3K-AKT pathways were found affected, with high diversity among patients. The phenomena from heterogeneous tumors all suggest that the initial genetic landscape of the tumors might influence the possible evolution of the cancer cells under therapeutic pressure.

The first observed efforts aimed to strengthen the inhibitory effect of BRAFi by involving dual inhibition affecting the MAPK pathway. This led to the often-applied combined therapy regimens in the present clinical practice. However, long-term success is still not possible in around 80% of the patients. A refinement in temporal application of the two drugs might be a plausible way to elongate a relapse-free period, as introduction of MEK inhibitors (MEKi) only when RAF resistance is already occurring delays the development of MEKi resistance [17].

Further development of BRAFi molecules lead to more selective binding and inhibition of the BRAF V600E protein. Pharmacokinetics could also be further improved, however, as shown in many studies, their inhibitory effects are simply bypassed; therefore, the refinement of the BRAFi molecules are not expected to be successful per se [18].

It was proposed that it can be more successful to target complex changes during the adaptation of melanoma to the BRAFi (+MEKi) treatment, but before progression, as early tolerance seems easier to control, as was the case with the application of HIV-protease inhibitor nelfinavir in combination with BRAFi or MEKi therapies, suppressing the PAX-MITF activation system [19].

Importantly, the reactivation of cell apoptosis can be an effective additional modality. As described, chemical reactivation of p53 proved to be highly efficient in restoring the applicability of vemurafenib after developed resistance, especially in cases when the PI3K-AKT pathway was affected in the cancer cell adaptation to initial treatment [15].

The aim of the present study is to establish a vemurafenib-resistant PDTX model in order to be able to investigate potential drug candidates, as well as identifying the underlying mechanisms driving the resistance in a complex model system, suggesting new potential targets for the management of BRAFi-resistant melanomas.

## 2. Materials and Methods

### 2.1. Chemicals

Vemurafenib, by the commercial name Zelboraf (Roche, Basel, Switzerland) pills were pulverized with a grinder and dissolved in sterile water to reach concentration of 10 mg/mL. Oligonucleotide primers were obtained from Sigma-Aldrich (Merck KgaA, Darmstadt, Germany).

### 2.2. PDTX Generation

KINETO Lab Ltd. provided the animal house facility to create, archive, and use PDTX models. In model generation, the fresh surgical samples are implanted into three 8-week-old NOD.Cg-*Prkdc^scid^ Il2rg^tm1Wjl^*/SzJ (NSG) mice (The Jackson Laboratory, Bar Harbor, ME, USA). Under anaesthetization, approximately 5–10 mm^3^ tumor tissue pieces were implanted subcutaneously under the right dorsal skin, and wounds were closed with Novosyn 4/0 surgical suture (B. Braun, Melsungen, Germany). Tumor growth was followed, and grown tumors were removed and cut to 5–10 mm^3^ pieces. A part of the samples was cryopreserved in the freezing solution of 50% Fetal Bovine Serum FB-1001, 39% DMEM LM-D1111, 1% Penicillin/Streptomycin LM-A4118, all from BioSera, Cholet, France; and 10% DMSO (D2650, Sigma, St. Louis, MO, USA), and slowly cooled to −80 °C using cell cooler, then transferred to liquid nitrogen after one day. Other than that, snap-frozen samples, FFPE-embedded samples are also routinely archived. At the animal house of the National Institute of Oncology, serial transplanting was carried out using 8- to 12-week-old NOD.CB17-Prkdcscid/NCrCrl (NOD-SCID) mice, obtained from Charles River Laboratories (Wilmington, MA, USA). The tumors are cultured up to not more than 10 generations of tumor-bearing mice in order to avoid genetic drift.

### 2.3. BRAF Genotyping

The original genotype of the source patient was determined in the clinical routine at the Department of Dermatology, Venereology and Dermatooncology, Semmelweis University, Budapest. Further identification of the mutant genotype was available from RNAseq data. BRAF V600E genotype was also confirmed by Western blot.

### 2.4. Animal Model of Vemurafenib Treatment

Tumor tissue was implanted into 8- to 12-week-old male NOD-SCID mice. When tumor sizes reached approximately 100 mm^3^, we initiated per os treatment with 100 mg/kg vemurafenib five times a week. Vemurafenib solution was stored at 4 °C for not more than a week, and vortexed before every use. When either the tumor sizes reached 1500–2000 mm^3^, or the animals reached the age of about 6 months (treatment groups), the tumors were removed, and spread as follows: from one tumor, the half of the tissue was serially transplanted into new young animals, while the rest of the tumors were snap-frozen within 10 min to ensure RNA and protein integrity, and kept at −80 °C for further use.

For the short-term experiment, we transplanted original PDTX tumors into 12-week-old male NOD-SCID mice. Following the same treatment protocol, we treated the animals daily for the duration of five days, and at the end of the fifth day, we removed the tumors, cut them into half, and one half was snap-frozen, while the other half was put into 4% paraformaldehyde for a routine FFPE embedding protocol.

### 2.5. RNA Isolation and RNAseq of Whole Genome mRNA

The relative RNA expression of *AGAP9*, *GZMB*, *GNRHR*, *CD27*, *CP*, *IFI27*, *GPR39*, *GSTA1*, *SLC15A3*, *FKBP1A-SDCBP2*, *PDGFA*, and *ABCB1* genes was determined using the RT-qPCR method. Total RNA was isolated from PDTX tumor samples using Trizol^®^ reagent (Life Technologies, Carlsbad, CA, USA) and Direct-zol RNA Miniprep kits (Zymo Research, Irvine, CA, USA). The concentration and purity of the extracted RNA samples were measured using a NanoDrop spectrophotometer (NanoDrop, Wilmington, DE, USA) at an absorbance of 260 nm and 280 nm. Synthesis of cDNA was performed in an Eppendorf 5331 Mastercycler Gradient thermocycler (Eppendorf, Enfield, CT, USA). 300 ng of total RNA was used according to the protocol of the used AMV-Reverse Transcription System (Promega, Madison, WI, USA). The cDNA samples were stored at 4 °C until further application. For real-time quantitative PCR (qPCR), the used primer sequences are summarized in Table 1. Measurements were carried out using SsoAdvanced Universal SYBR^®^ Green Supermix assay (Bio-Rad, Hercules, CA, USA) with a CFX96 Touch Real-Time PCR Detection System (Bio-Rad). Relative expressions were determined by normalizing to the expression levels of RPLP0. All measurements were included in one gene study in order to maintain common threshold values.

The prepared total RNA samples were sent to Xenovea Ltd. (Szeged, Hungary) for transcriptome analysis. The RNA concentration was determined by using the Qubit RNA HS Assay Kit on the Qubit 3.0 Fluorometer (Thermo Fisher Scientific, Waltham, MA, USA). Quality control was assessed by Labchip GX Touch HT instrument on DNA 5K/RNA/CZE Chip (Perkin Elmer, Waltham, MA, USA) with RNA Pico Sensitivity Assay Reagents (Perkin Elmer). NextFlex PolyA beads 2.0 kit and NextFlex Rapid Directional RNA-seq Kit 2.0 with UDIs (Perkin Elmer) were used for mRNA capture and strand-specific library preparation. The library quantities were measured by Quant-iT 1× dsDNA HS Assay kit (Thermo Fisher) with Fluostar Omega (BMG Labtech, Ortenberg, Germany). The fragment size distribution of the libraries was determined by capillary electrophoresis on Labchip GX Touch Nucleic Acid Analyzer on XMark HT chip by using DNA NGS 3k Assay kit (Perkin Elmer). Pooled libraries were sequenced with 50 M 150 bp paired-end reads on NovaSeq 6000 platform (Illumina, San Diego, CA, USA).

### 2.6. Bioinformatic Analysis of RNAseq Data

The sequencing output reads were adapted and 5′-3′ end trimmed for min. Q30 score, and the reads were filtered for minimum mean quality (Q30) and 100 nt length with Illumina BaseSpace FASTQ Toolkit App (v.2.2.5), and polyA trimmed with cutadapt (v3.5) [20] with parameters—a “A{100}” A “A{100}”. Trimmed reads were mapped to reference genome hg38, with spliced-aligner hisat2 (v.2.1.0) [21]. The resulting mapping files were converted to BAM, collated, sored, and a duplicate marked with samtools (1.15.1) [22]. Transcripts were quantified using StringTie (v2.1.1) [23], and a count matrix was generated with supplied prepDE.py script. Differential expression analysis was performed using the R package limma (3.48.0), limma-voom method, with trimmed mean of M values (TMM) normalization for library size [24,25,26,27]. Genes with low expression were filtered prior to analysis (filtering criteria: min. count 3 in min. 3 samples). The *p*-value was adjusted using the Benjamini and Hochberg method [28]. Genes with log_2_FC ≥ 1.5, and FDR < 0.05 were considered upregulated, similarly log_2_FC ≤ −1.5, and FDR < 0.05 were considered downregulated.

### 2.7. Alternative Splicing Detection

The method used for alternative splicing detection was reported previously [29]. From previously obtained cDNA samples, we used 4 μL templates to perform a 20 μL reaction using GoTaq Green polymerase reagent and the proper BRAF primer pairs (exon 3–9 and exon 8–9, see also Table 1). The PCR program included 40 cycles of polymerase steps (95 °C for 30 s, 52 °C for 30 s, 72 °C for 60 s). The PCR products were then put to a 1% agarose gel electrophoresis, containing 1:20,000 ECO Safe Nucleic Acid staining solution for visualization. The gel was documented using iBright 750 equipment (Life Technologies).

### 2.8. Western Blot

Protein expressions in PDTX samples were carried out as previously detailed [30], with the following modifications: from flash-frozen tissue samples, we used frozen grinding (sterile manual grinder, frozen with a continuous addition of liquid nitrogen, pulverizing the sample), and the resulting tissue powder was immediately put to ice with 200 µL of Mg^2+^ lysis buffer containing aprotinin, leupeptin, and phosphatase inhibitor PMSF. Gel electrophoresis and Western blotting were performed according to the manufacturer’s protocol in 1 mm 10% SDS polyacrylamide gels, and the Turbo-Blot semidry blotting system (Bio-Rad). Membranes were blocked for 1 h with 5% nonfat dry milk dissolved in TBS-Tween, then incubated with the primary antibodies overnight (4 °C). After washing and secondary labeling (1 h, RT), HRP signals were visualized with the WesternBright ECL system (Advansta, San Jose, CA, USA), and documented with the iBright 750 documentation system (Thermo Fisher, Waltham, MA, USA).

Antibodies used: AKT: #4691; p-AKT: #13038; mTOR: #2983; p-mTOR: #5536; p44/42 (ERK1/2): #4695; p-p44/42 (p-ERK1/2) #4370; PDGFRB: #3169; p-MEK 1/2: #9154; p-c-RAF: #9421; p53: #9282; GAPDH: #2118, all from Cell Signaling Technologies (Danvers, MA, USA), CD27 (ab175403), IFI27 (ab171919), both from Abcam PLC (Cambridge, UK), diluted and applied according to the manufacturer’s instructions. An anti-BRAF V600E mutation-specific antibody (clone VE1) was obtained from Ventana Medical Systems (Oro Valley, AZ, USA) and diluted in nonfat dry milk—TBS tween solution at a concentration of 1.5 μg/mL. Secondary anti-mouse (#7076) or anti-rabbit (#7074) IgG antibodies conjugated with HRP were purchased from Cell Signaling and used according to their manual.

### 2.9. Immunohistochemistry

The slicing, preparation, and staining of the samples were carried out following standard protocol [31] with the rabbit anti-human monoclonal antibody against p53 (#9282), ABCB1 (#13342), both from Cell Signaling, or specific antibodies against GPR39 (ab229648) from Abcam, GnRHR (19950-I-AP) from ProteinTech, used in 1:200 dilution. Rat anti-mouse F4/80 (MCAP497) from AbD Serotec (Oxford, UK) was used in 1:100 dilution. After overnight staining and wash, the appropriate host-specific HRP-conjugated secondary antibody was applied and visualized. Whole tumor sections were scanned using a Panoramic 250 Flash scanner (3D Histech Ltd., Budapest, Hungary). Obtained images of non-necrotic parts of the samples were analyzed using the positive-negative cell counting function of the QuPath 0.4.3. software [32].

### 2.10. Data Processing and Statistical Analyses

Tumor growth data was documented and graphed using Microsoft Excel. The relative expression levels in qPCR were evaluated by the CFX Maestro 1.1 software package (Bio-Rad), using the built-in ANOVA analysis, while graphs were visualized in Microsoft Excel as well as for immunohistochemistry data. The statistical methods for bioinformatic analysis are detailed in the dedicated chapter of the methods. Immunohistochemistry data was visualized using Microsoft excel, and statistical significance was calculated using Student’s *t*-test (two groups) or one-way ANOVA (more groups).

## 3. Results

### 3.1. PDTX Growth and Vemurafenib Resistance In Vivo

Our subcutaneous PDTX model of malignant melanoma was treated on long-term to follow possible acquired drug resistance. As the results show, the treatment was effective to achieve stable disease for over two months of treatment. In the following generations, tumor growth was accelerating in a treatment time-dependent fashion. By generations 3–6, we saw developed resistance (Figure 1).

Initially, two groups (treated with vemurafenib or water per os) were started; then, for further development of resistance, the treated tumors were serially transplanted and sampled. In the sixth generation, a treatment-withdrawal group was started in order to detect any reversal of the resistance.

### 3.2. RNAseq Results Reveal Differentially Expressed Genes over Vemurafenib Treatment

From each mouse, flash-frozen samples were prepared and used for genomic RNA generation. Based on tumor growth data, a total of nine samples were chosen from generations 1 (untreated), 3, and 5 (three animals from each group). The transcriptomic analysis by mRNAseq showed a clear grouping of biological groups in the primary component analysis, and provided a list of 3991 significantly differentially expressed genes in treated tumors (Figure 2), among which 434 genes were significantly altered both in short and long treatment groups.

### 3.3. Detailed Analysis of Highlighted Genes of Interest by qPCR

After filtering the genes commonly changed in generation 3 and generation 5 tumors, we created a list of 12 genes which had at least 5-fold difference in both treated groups, and furthermore were reported as prognostic factors in any cancer types, according to the ProteinAtlas database. Additionally, we included 2 more genes to our panel: gonadotropin-releasing hormone receptor (*GNRHR*), a differentially expressed gene, which is not reported as prognostic, but our previous work showed its possible targetability in cancer, and *CD27*, which did not meet the fold-change criteria but was reported to have positive prognostic effect in melanoma patients. Additionally, being involved in many resistance mechanisms, we double-checked the expression levels of the three major ATP-binding cassette subfamily members, *ABCB1*, *ABCG2*, and *ABCC1* (Table 2).

The measurements did not find measurable quantities of *ACBD7-DCLRE1CP1* fusion gene, *TNNT3*, *ZNF141*, *ABCG2*, and *ABCC1*. The other gene expressions were measured in all the available samples (*N* = 52) of the six generations of treatment, control, or withdrawal samples. All the expression patterns are summarized in Figure 3.

We found that the expression of *GSTA1*, *FKBP1A-SDCBP2*, *GZMB*, *AGAP9*, *CP*, and *GNRHR* showed strongly elevated expression in G3 samples but lacked a consistent, resistance-dependent expression pattern (Figure 3). While *GPR39* shared the same peak, other treated samples also showed about a 10-fold upregulation. Oddly, *GPR39* was found downregulated in the primary search, but in the qPCR results, it was found upregulated, raising the importance of different detection methods. *PDGFA* was found to be mildly upregulated, raising the possible role of PDGF-PDGFR signaling in the acquired resistance. A consistent, strong downregulation of *IFI27* was recorded, which can suggest its role in the formation of vemurafenib resistance in our PDTX model. Among multidrug resistance genes, *ABCB1* was measured to reach about a 15-fold increase, with an extra peak in G3 samples. This effect was not diminished in the withdrawal group (Figure 4).

The results are indicating previously unnamed genes that are possibly involved in acquired vemurafenib resistance.

### 3.4. Protein Expression of Differentially Expressed Genes

To examine whether the expression pattern of the investigated genes is accordingly translated to protein quantities, we performed immunohistochemistry and quantitative image analysis of formalin-fixed paraffin embedded tissue sections. The antibody F4/80 marked macrophages, the same cell type that SLC15A3 is expected to be expressed on. Large variation among samples in the same group hurdled significant statistical power upon evaluation of the results, but trends were observed upon quantification. The expression levels of GPR39 were rising after treatment, especially from the third generation samples. On the other hand, *ABCB1* gene product P-glycoprotein (P-gp) and F4/80 macrophage marker expressions were elevated after the first months of treatment, followed by a normalization, despite of the maintained high expression. This hypothesizes some feedback mechanism or evolutionary cost of overexpression on longer term. GNRHR levels were also measured, and found high levels in all samples (making it a plausible special targeting point) but without big differences between sample groups (Figure 5A,B).

Two proteins unavailable for immunohistochemical analysis were subjected to Western blot: CD27 and IFI27 were both present in all treatment-naïve (G1) and resistant (G5) samples. However, in-group variance was high; therefore no significant changes could be proved (Figure 5C).

### 3.5. Analysis of Known Resistance Mechanisms in the Present Resistance Model

As mentioned in Introduction, numerous different mechanisms are described to cause experimental or clinical resistance. We chose to investigate these mechanisms in order to delineate the effects found in the transcriptomic data from the possible confounding effects of another resistance mechanisms.

Regarding the known SNPs reported in the literature, we used the RNAseq data to capture allelic differences. In all nine samples, we found a uniform pattern of the following: all samples were heterozygous for BRAF V600E, without frequency change in the transcripts. The tumors were homozygous for RAC1 N92K mutation, and it was the case in every untreated and treated sample; therefore, no genotype change was present due to the treatment.

All samples were found wild type for the following mutational spots: NRAS Q61; KIT L576 and K642; RAC1 P29 and MAP2K2 Q60.

For the alternative splicing detection, we used reverse transcription, cDNA PCR, and gel electrophoresis to show products which are available only in the wild type (exon 8–9 probe) and both in wild type and exon 4–8 deleted forms but with different sized products (exon 3–9 probe). All 9 samples examined in the NGS analysis were found to carry full length *BRAF* only (Figure 6).

The possible activation of different pathways, amplification of BRAF, or activation of CRAF were investigated, and no such accumulations or activation of these proteins that could show any of the given mechanisms of vemurafenib resistance were found. Furthermore, p53 accumulation inducing apoptosis was not found (Figure 7).

### 3.6. Analysis of Apoptotic Activity in Control and Treated PDTX Tumors

As avoiding apoptosis was also named as a possible outcome of the acquired genetic changes following vemurafenib treatment, we wanted to show whether it is the case in our experimental model. To do this, first, we checked the p53 protein levels in the Western blot and found that apoptotic activity was similar in treatment naïve and resistant samples. However, as in theory this could be the combined result of an elevated apoptotic activity on vemurafenib, and the decreased activity due to resistance, we ran a short treatment experiment from the initial samples. After 5 days of treatment, we collected the tumors, and we performed immunohistochemical staining against human p53 on FFPE sample tissue slices. We analyzed three large areas, which were not necrotic to investigate p53 nuclear signals. The results did not show significant elevation of the p53-positive apoptotic cells, while resistance was not developed in this short time span (Figure 8).

## 4. Discussion

Malignant melanoma, given its highly metastatic behavior, needs an effective therapeutic approach for systemic treatment. The cytotoxic therapies do not offer a successful solution [33], while targeted therapies [34] and immunotherapies [35] might be more successful. The common *BRAF* V600E mutation is druggable, and selective inhibition of the altered BRAF protein has good antitumor effects [36]. Still, therapy resistance occurs in the vast majority of cases [37]. Genetic and protein-level studies found many different mechanisms in cell line models in vitro and in vivo, and several PDTX and clinical studies were performed as well. Common upregulation of downstream signaling (MAPK pathway) led to the application of double inhibition of BRAF V600E and ERK 1/2 to achieve a better outcome and elongated remission [11,12]. However, the combined therapy has not resolved the plasticity of cancer cells, leading to therapy failure [38].

In our present study, we introduced a patient-derived tumor xenograft model from a *BRAF* V600E mutant melanoma patient, with high intratumoral heterogeneity and microenvironment characteristics. With long-term vemurafenib treatment, we successfully developed a therapy-resistant version of the tumor. Our aim was to confirm any existing mechanisms in our model, as well as to try to detect previously unknown changes that can cause in vivo resistance. As we checked the detailed analysis of RNAseq measurements, we did not find any existing theories [39] to fit and focused on differentially expressed genes. In addition, rarely focused on in melanoma models, the multidrug transporter ABCB1 was also investigated, since it is capable of the recognition and export of the inhibitor molecule from the cells [40].

Our gene set was further investigated on 52 samples, and it resulted in a systematic upregulation of several genes. GPR39 is a G-protein coupled receptor, interacting with zinc, and promoting colonocyte survival. Its overexpression was reported in colon cancer as a factor for poor prognosis [41] but has not yet been investigated in melanoma. Its role in melanoma should be further investigated. CD27 is found on the surface of immune cells, such as activated T or B cells, and NK cells. Since in NOD-SCID mice T or B cell functions are diminished, we suspect that the infiltration of NK cells can be responsible for the phenomenon, as the model is not truly NK-deficient [42]. This phenomenon could be exploited by the stimulation of the present NK cells. Similarly, SLC15A3 is immune-cell related, mostly present on macrophages, having a role in inflammatory responses [43], which, in our case, can be driven by the therapy itself. *IFI27* (interferon alpha inducible protein 27) was the only confirmed downregulated gene in the present study, interestingly exhibiting the most solid expression pattern among all the generations of mice/samples. Its stabilization in esophageal cancer leads to tumor progression [44] and angiogenesis, while in breast cancer, metastases showed systematic downregulation of *IFI27* [45]. This somewhat contradictory background needs specific investigation of the plasma protein in melanoma to reveal its possible therapeutic values. PDGFA overexpression is linked to previously published data [46]. However, further analysis of the downstream signaling did not underlie this theory. At the same time, PDGFR inhibitory therapies (such as imatinib) are possible candidates to be tested in further studies about whether they can improve the BRAFi effect. *ABCB1* and its infamous product P-glycoprotein are well known as determinants of multidrug-resistance [47], maintaining intensive efflux of a large variety of drugs from the cells, causing treatment inefficiency. Indeed, it is reported that vemurafenib is a substrate of P-glycoprotein and BCRP (breast cancer resistance protein, ABCG2); therefore, *ABCB1* overexpression in the cancer cells can simply avoid intracellular accumulation and activity of vemurafenib [48]. Unfortunately, P-glycoprotein inhibitors are not applicable in vivo [49], but the development and the use of BRAF inhibitors not recognized by the transporter could become an effective alternative. Of note, the MEK inhibitor trametinib applied in combination therapy is also a proven substrate of the transporter [50].

The expression changes recorded were followed up on the protein level, and we found the interesting pattern that both P-glycoprotein and macrophage marker F4/80 were induced in some generations but normalized at generation 5. This might mean that while they can confer to initial resistance and augment survival, new strategies can be developed later, and their fitness-cost pushes to phenotype with less protein (although RNA expression remains high). Of contrast, the expression of GPR39 was measured to get elevated only at later stages. The exact function and cooperation of these proteins in developing resistance is yet unknown.

In comparison to other studies conducted in a similar setup, we found no evidence of the common resistance mechanisms including BRAF or CRAF activation, any enhancement in PDGFR expression, AKT-mTOR expression or phosphorylation levels, MAP2K2, p44/42 expression, or phosphorylation. Although this phenomenon is not contradictory, it widens the focus to possible alternative strategies of cancer cells to evade drug efficacy. To further acknowledge, the resistant tumors’ unchanged activity in ERK 1/2 activation means that the BRAFi therapy was no longer successful in silencing the main proliferative pathway involved, as proved in the clinical practice [51].

Alternative resistance mechanisms are reported elsewhere, for example, overexpression of MCL-1 [52] or the activation of PAX/MITF axis [19], DUSP suppression [53], ACOX1-mediated fatty acid oxidation activation [54]; however, in our model, these changes were not detectable.

It is of importance that in the case of several markers, we observed considerable variability between different animals. This raises the importance of intertumoral [55] and intratumoral [8,56] heterogeneity in terms of resistance development, which could occur parallel in different cell populations, as raised by other studies as well.

## 5. Conclusions

In summary, our model provided another prominent model of BRAFi-driven acquired resistance in malignant melanoma. The genetic analysis showed distinct mechanisms in these tumors, altering from commonly reported ways of therapy failure. The found alterations can be addressed, and for promising drug candidates, our resistant tumor model can serve as a proof-of-concept experimental platform in order to find new ways to control melanoma long term. Based on our study, new potential targets emerged that can be addressed by combination therapies in a subset of melanoma patients in the future.

## Figures and Tables

**Figure 1 cells-12-01919-f001:**
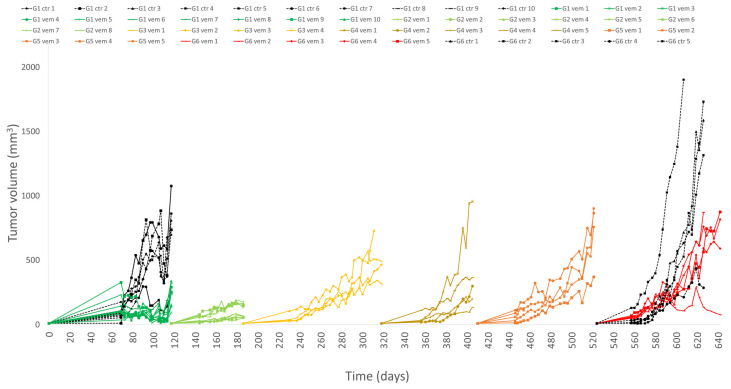
Tumor growth of melanoma PDTX model over 6 generations of mice.

**Figure 2 cells-12-01919-f002:**
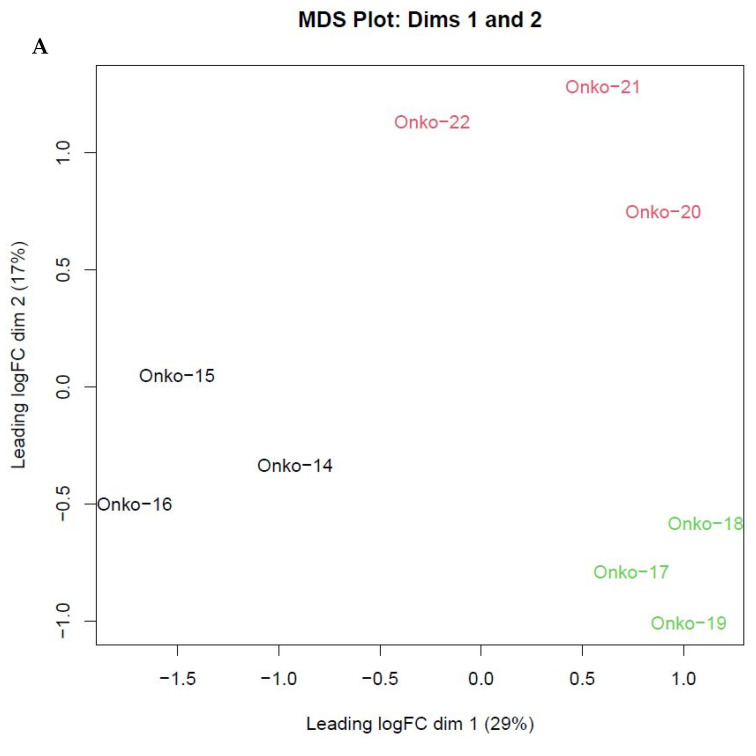
Gene expression patterns in untreated and treated BRAF V600E melanoma PDTX model. (**A**) The variance among samples was analyzed in primary component analysis, showing distinct groups of the untreated (black), generation 3 (green), and generation 5 (red) samples. (**B**) Expression pattern of 3991 differentially expressed RNAs versus the untreated samples. The *x*-axis represents expression in generation 5, and the *y*-axis the expression in generation 3 tumors.

**Figure 3 cells-12-01919-f003:**
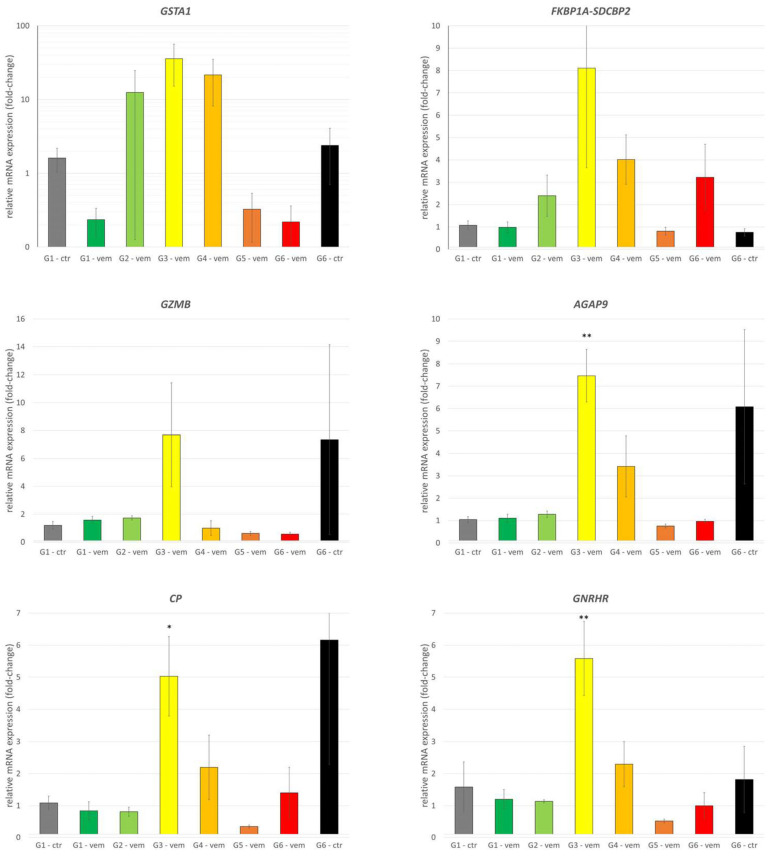
Relative mRNA expression changes of melanoma PDTX tissue upon vemurafenib treatment. All panels show relative expressions, normalized to control *RPLP0*. All measurements were analyzed in a common study to ensure the common threshold values for cT determination. Gene expressions (*GSTA1*, *FKBP1A-SDCBP2*, *GZMB*, *AGAP9*, *CP*, *GNRHR*) that share an expression peak in G3 samples but are not consistently changed. *: *p* < 0.05; **: *p* < 0.01.

**Figure 4 cells-12-01919-f004:**
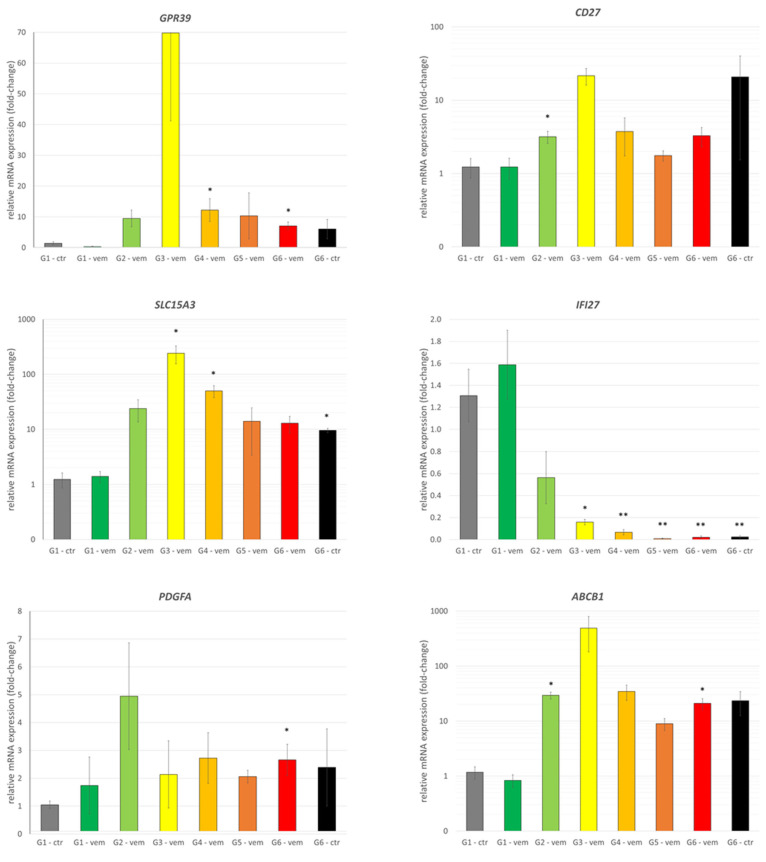
Relative mRNA expression changes of melanoma PDTX tissue upon vemurafenib treatment. All panels show relative expressions, normalized to control *RPLP0*. All measurements were analyzed in a common study to ensure the common threshold values for cT determination. Genes systematically upregulated (*GPR39*, *CD27*, *SLC15A3*, *PDGFA*, *ABCB1*) or downregulated (*IFI27*) after vemurafenib treatment. *: *p* < 0.05; **: *p* < 0.01.

**Figure 5 cells-12-01919-f005:**
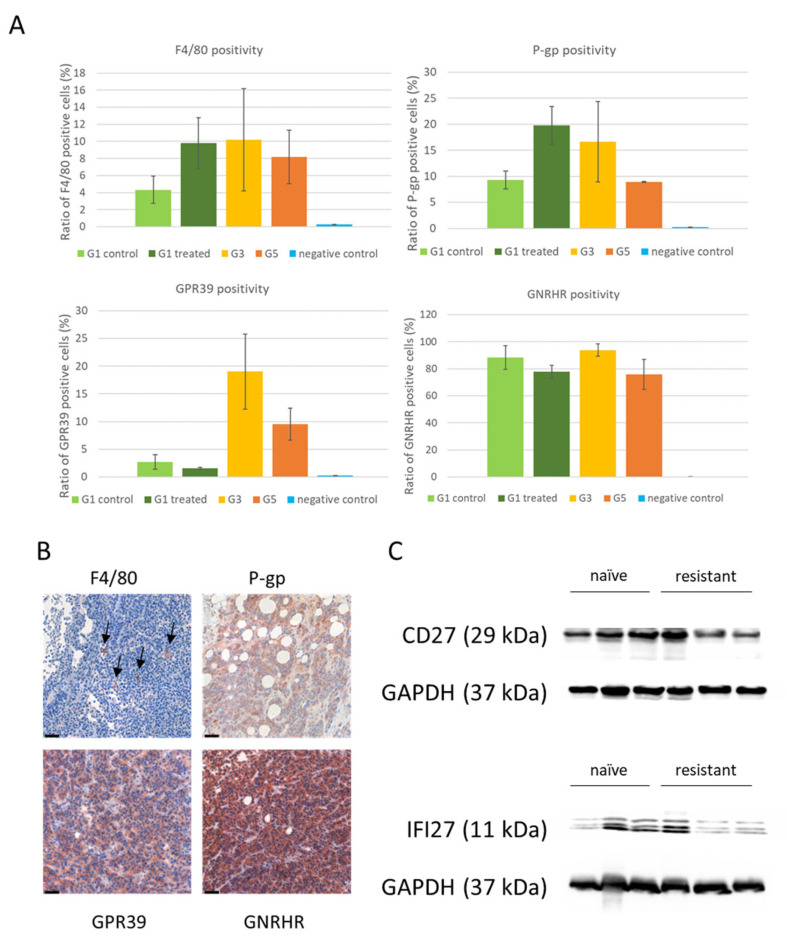
Protein expression patterns upon vemurafenib treatment in melanoma. (**A**) Quantitative analysis of positive and negative cells from whole-section scanned images. *N* = 3 tumors in each groups. (**B**) Representative immunohistochemical staining pictures of P-gp, GPR39, F4/80, and GNRHR proteins. Black arrows indicate F4/80-positive macrophages. Scale bars: 50 µm. (**C**) Western blot images showing protein level changes of CD27 and IFI27. Three control and three treated animal samples were measured.

**Figure 6 cells-12-01919-f006:**
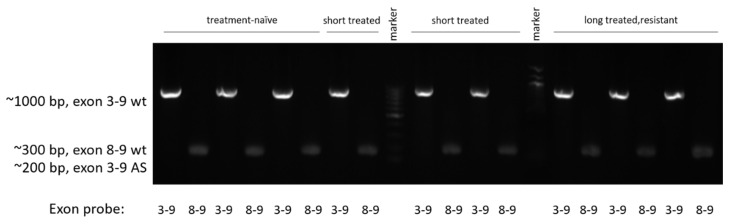
All samples carried full length *BRAF* mRNA. Both with primers on exons 8 and 9 (300 bp product) and primers 3–9 (1000 bp product), we identified the full length *BRAF* mRNA but not the exon 4–8 deleted variant, which could have been visualized in the exon 3–9 probe, at 200 bp size.

**Figure 7 cells-12-01919-f007:**
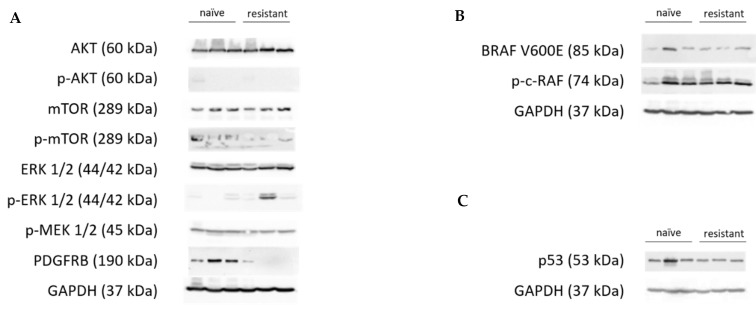
Protein levels and activation of different signaling pathway proteins, BRAF V600E, and activated CRAF, and p53 expression. The figure shows three parallel animals’ tumor samples from the treatment-naïve group and from generation 5. GAPDH was used as an endogenous control to ensure equal protein levels of every sample. (**A**) Neither AKT, mTOR, MAP2K1/2 (MEK 1/2), or P44/42 (ERK 1/2) was found to be systematically activated, though the ERK 1/2 expression was found elevated in one of the three resistant tumors. Moreover, PDGFRB, which showed a 4-times elevation in RNAseq data, was shown not to be overexpressed on the protein level. (**B**) BRAF V600E and p-c-RAF expressions remained unchanged on the vemurafenib treatment. (**C**) p53 accumulation is similar in both the naïve and resistant (and treated) samples.

**Figure 8 cells-12-01919-f008:**
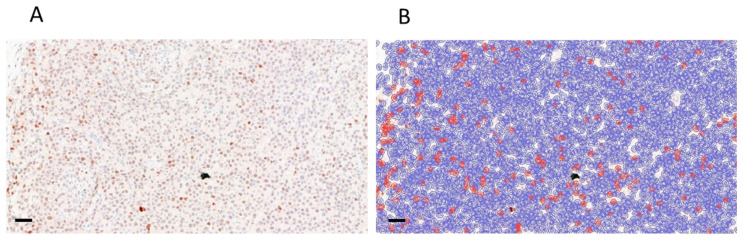
Quantitative analysis of apoptosis by counting p53 positive cells. (**A**) We used the scanned tissue slides to analyze three view areas per slide, examining in total above 7000 cells in each tumor slide. (**B**) We trained QuPath 0.4.3. software to identify negative and positive cells (red and blue marks, respectively) under visual control in order to use the best image analysis parameters. Scale bars: 50 µm. (**C**) p53-positive apoptotic cell ratio in control (*N* = 4) and treated (*N* = 3) tumors. (**D**) Average p53-positive apoptotic cell ratio in control and treated tumors.

**Table 1 cells-12-01919-t001:** Oligonucleotide primers used in the study.

Gene	Forward Primer	Reverse Primer
*AGAP9*	CCACCACTGATGAGGACCTG	ATGACGTCCACCCCGTCA
*GZMB*	CCCTGGGAAAACACTCACACA	GCACAACTCAATGGTACTGTCG
*GNRHR*	TGTCTGGAAAGATCCGAGTGA	AGGTTGGCTAAGGTCAGATGTT
*CD27*	TCAGCAACTGGGCACAGAAA	GGATCACACTGAGCAGCCTT
*CP*	GGGCCATCTACCCTGATAACA	TTAAAGGTCCGATGAGTCCTGA
*IFI27*	TGCTCTCACCTCATCAGCAGT	CACAACTCCTCCAATCACAACT
*GPR39*	TGTCCCCGAGTTTGAGGTG	GAAGGCCCATCACGAAGATGA
*GSTA1*	CTGCCCGTATGTCCACCTG	AGCTCCTCGACGTAGTAGAGA
*SLC15A3*	TGGCGTTTATTCAGCAGAACA	TCTCTGGCCGAGTGTCGTT
*FKBP1A-SDCBP2*	CACTACCCTGCACTGAGCTG	CATGACGTCCACCCCGTCAG
*PDGFA*	GCAAGACCAGGACGGTCATTT	GGCACTTGACACTGCTCGT
*ABCB1*	TTGCTGCTTACATTCAGGTTTCA	AGCCTATCTCCTGTCGCATTA
*RPLP0*	AGCCCAGAACACTGGTCTC	ACTCAGGATTTCAATGGTGCC
*BRAF* exon 3	AGCAAGCTAGATGCACTCCA	---
*BRAF* exon 8	CCAAATTCTCACCAGTCCGT	---
*BRAF* exon 9	---	ACCACGAAATCCTTGGTCTC

**Table 2 cells-12-01919-t002:** Upregulated, downregulated, and MDR genes chosen for qPCR analyses.

**Upregulated** **genes**	**Fold-change:** generation 5/control	**Fold-change:** generation 3/control	**Positive prognostic value** (ProteinAtlas)	**Negative prognostic value** (ProteinAtlas)
*AGAP9*	59.54	102.30	UC	RCC; CRC
*FKBP1A-SDCBP2*	25.83	16.11	---	HCC; RCC
*GZMB*	8.21	9.49	EC; BC	RCC
*CP*	9.49	7.60	---	RCC
*GNRHR*	6.41	8.94	---	---
*ACBD7-DCLRE1CP1*	7.80	5.77	HNSCC; OC	RCC
*CD27*	3.04	2.86	Melanoma malignum; RCC	HNSCC; CC; EC

**Downregulated** **genes**	**Fold-change:** control/generation 5	**Fold-change:** control/generation 3	**Positive prognostic value** (ProteinAtlas)	**Negative prognostic value** (ProteinAtlas)
*GPR39*	22.97	32.52	---	Pancreas
*GSTA1*	18.17	16.61	NSCLC	RCC
*TNNT3*	16.55	15.13	CC	---
*SLC15A3*	15.78	7.71	CC	---
*IFI27*	49.45	7.11	OC	---
*ZNF141*	29.13	6.88	UC	---
*PDGFA*	5.13	5.31	---	glioma, HNSCC, UC

**MDR** **genes**	**Fold-change:** generation 5/control	**Fold-change:** generation 3/control	**Positive prognostic value** (ProteinAtlas)	**Negative prognostic value** (ProteinAtlas)
*ABCB1*	n.s.	n.s.	RCC; Pancreas	---
*ABCG2*	n.s.	n.s.	---	---
*ABCC1*	n.s.	n.s.	---	RCC; HCC

Abbreviations: UC: urothelial carcinoma; RCC: renal cell carcinoma; CRC: colorectal cancer; HCC: hepatocellular carcinoma; EC: endometrial cancer; BC: breast cancer; HNSCC: squamous cell carcinomas of the head and neck region; OC: ovarian cancer; CC: cervical cancer; NSCLC: non-small cell lung cancer; n.s.: not significant.

## Data Availability

All data acquired in experiments are either presented in this study or are available at the authors for reasonable request. The entire RNAseq differentially expressed genes list with change of expression and adjusted significance levels are included as a Appendix A.

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
