# Peer review of "Evolving Acquired Vemurafenib Resistance in a BRAF V600E Mutant Melanoma PDTX Model to Reveal New Potential Targets"

_cells, 2023, doi:10.3390/cells12141919_

Round 1

Reviewer 1 Report

This manuscript proposes acquired vemurafenib-resistance in a BRAF V600E mutant melanoma PDTX model to reveal new potential targets.  The detailed mechanisms underlying the Vemurafenib-Resistance were studied intensively and explained. The experiments were carried out accurately and underline the given conclusions. Results are discussed in accordance to the current respectable references, leading to the conclusions of significance for the modern treatment of melanoma patients.   The current manuscript has some studies that need to be addressed.

There are some of the studies that can be considered.

1) The authors also need to show that genes upregulated by RNA seq in PDTX vemurafenib-resistance model  are translated in protein expression level.

2) The authors could include immunoblots of naïve and resistant models (based on RNA seq analysis) so that you know that protein is playing in resistance mechanism as they showed in the manuscript only the AKT and MAPK protein which did not play resistance mechanism in this PDTX model.

The studies documented in this manuscript are interesting and novel since it did not show common pathways that were upregulated (reactivation of MAPK or parallel pathway PI3K-AKT) upon vemurafenib-resistance. This PDTX model has shown novel potential therapeutic targets. I recommend acceptance with minor revisions.

Author Response

We are thankful for the reviewer's expert opinion. We went through the points and added additional results to the manuscript. The detailed answers can be found in the attached Response letter, please see the attachment.

Reviewer 2 Report

The authors of the manuscript developed a patient-derived tumor xenograft model (PDTX model) to screen melanoma drug resistance in mice. According to my understanding, the PDX model was established to investigate new molecular targets in BRAF-resistant melanoma. The authors used a patient-derived BRAFV600E mutant melanoma to develop the PDX model. Different techniques such as bioinformatics, alternative splicing, and immunohistochemistry were employed in the study.

The investigation is contemporary as the literature has demonstrated the applicability of PDX models in the development of anti-cancer drugs, co-clinical trials, personalized medicine, immunotherapy, and PDX biobanks.

However, many points should be addressed.

1. Introduction is a puzzle. Very difficult to understand. It is stated about BRAF inhibitors, then regarding MAPK inhibition. In sequence, the authors try to show a connection between PDX models and BRAF inhibition. What is the rationale of the investigation?

2. The authors do not show clearly the goal of the manuscript. 

3. The language is so confusing. It is really difficult to understand the manuscript.

4. Carefully missing. Not only many grammar errors but also catalog numbers and parentheses (page 5, line216 – 218), lowercase letters at the beginning of sentences (page 1, line 39), capital letters in the middle of the text (page 2, line 58), missing references (page 5, line 195).

5. Material and Methods are disorganized. In the 2.1 Chemicals is cited only vemurafenib. Why were the other chemicals used in the study not cited?

6. Statistical Analysis does not cite correct tests to be used in the study. 

7. Figures are not properly cited in the results.

8. In the results, Table 1 shows a list of upregulated and/or downregulated genes in different types of cancer. Among 14 genes, only one presents a positive prognostic value for melanoma. What is the rationale for the 13 genes unrelated to melanoma? 

9. Figure 3 shows relative mRNA expression changes upon vemurafenib treatment. However, it is not clear where mRNA was obtained. In addition, the figure presents many graphics with very little letters on the x and y-axis. It is difficult to visualize. 

10. Figure legends are not clearly describing the respective results.

11. Because there is no logical rationale, discussions show several genes involved in different types of cancer, and some speculations are made.

12. The manuscript concludes that a successfull BRAF-resistant PDX-model was built and could be a usefull tool to investigate promising drug candidates. Unfortunately, data presented in the manuscrip in the manner it is presented did not support the conclusion.

The quality of English should be improved.

Author Response

(The authors gave the same response as above.)

Round 2

Reviewer 2 Report

Dear authors,

thank you for the revised version. The quality and understanding of the manuscript has been improved.

Best regards,